# Ultrasensitive Silicon Photonic Refractive Index Sensor Based on Hybrid Double Slot Subwavelength Grating Microring Resonator

**DOI:** 10.3390/s24061929

**Published:** 2024-03-17

**Authors:** Kaiwei Lu, Beiju Huang, Xiaoqing Lv, Zan Zhang, Zhengtai Ma

**Affiliations:** 1Key Laboratory of Optoelectronic Materials and Devices, Institute of Semiconductors, Chinese Academy of Sciences, Beijing 100083, China; lukaiwei@semi.ac.cn (K.L.); lvxiaoqing284@semi.ac.cn (X.L.); mazhengtai@semi.ac.cn (Z.M.); 2University of Chinese Academy of Sciences (UCAS), Beijing 100049, China; 3College of Materials Science and Opto-Electronic Technology, University of Chinese Academy of Sciences (UCAS), Beijing 100049, China; 4Suzhou Institute of Microelectronics and Optoelectronics Integration, Suzhou 215213, China; 5School of Electronics and Control Engineering, Chang’an University, Xi’an 710064, China; z.zhang@chd.edu.cn

**Keywords:** silicon photonics, refractive index sensor, hybrid double slot subwavelength grating waveguide, ultra-high sensitivity

## Abstract

Silicon photonic-based refractive index sensors are of great value in the detection of gases, biological and chemical substances. Among them, microring resonators are the most promising due to their compact size and narrow Lorentzian-shaped spectrum. The electric field in a subwavelength grating waveguide (SWG) is essentially confined in the low-refractive index dielectric, favoring enhanced analyte-photon interactions, which represents higher sensitivity. However, it is very challenging to further significantly improve the sensitivity of SWG ring resonator refractive index sensors. Here, a hybrid waveguide blocks double slot subwavelength grating microring resonator (HDSSWG-MRR) refractive index sensor operating in a water refractive index environment is proposed. By designing a new waveguide structure, a sensitivity of up to 1005 nm/RIU has been achieved, which is 182 nm/RIU higher than the currently highest sensitivity silicon photonic micro ring refractive index sensor. Meanwhile, utilizing a unique waveguide structure, a Q of 22,429 was achieved and a low limit of detection of 6.86 × 10^−5^ RIU was calculated.

## 1. Introduction

In recent years, optical sensors based on label-free detection technology have been widely used in medical diagnosis and environmental monitoring due to their low cost, fast response, and high sensitivity [1,2,3,4,5]. Among, the biosensor based on surface plasmon resonance (SPR) is the most well-known and widely used label free optical biological analysis tool since its successful commercialization [6]. However, SPR based biosensors still face challenges such as system miniaturization and high detection environment costs [7].

In label-free optical sensors, sensors based on silicon photon technology are widely explored for manufacturing integrated optical sensors at the μm or nm scale [8]. Large scale integrated photonic circuits (PIC) based on optical waveguides can integrate passive photonic devices with various functions on a single chip, as well as active devices such as lasers and photodetectors [9,10]. Meanwhile, the low absorption and high refractive index contrast of silicon in the near infrared (near-IR) band make the device size more compact [11]. Finally, silicon on insulator (SOI), which has been used as the main platform for implementing PIC, is compatible with complementary metal oxide semiconductor (CMOS) technology, which means that high integration devices can be mass-produced at low cost. The above advantages all demonstrate the enormous application prospects of label free biosensors based on silicon photon technology in portable sensors with low-cost efficiency and miniaturization. In integrated optical sensors, sensing is usually carried out by utilizing the interaction between the evanescent field leaked in the optical waveguide and environmental substances. In this process, changes in the concentration and content of the material detected by the cladding cause changes in the absorption coefficient or effective refractive index of the optical waveguide, which ultimately leads to changes in the output spectrum or light intensity of the sensor device.

The type of refractive index sensor based on silicon photonics is also a functional device that can resonate, interfere, and be absorbed by light waves, including microdisk resonator sensors [12,13,14], microring resonators (MRR) sensors [15,16,17,18], Mach-Zehnder interferometer (MZI) Sensors [19,20,21], multimode waveguide interference coupler sensors [22,23], photonic crystal cavities [24,25]. Or it could be a combination of MZI-MRR, MRR-MRR, and MZI-MZI, based on the Vernier effect that significantly increases sensitivity [26,27,28,29]. Especially, microring resonators have attracted widespread attention due to their small size and sharp Lorentz resonance peaks [30,31]. Sharp resonant peaks indicate a high Q factor, which reduces the detection limit of refractive index sensors. For MRR, the essence is to design a unique waveguide structure to fully release the interaction between the evanescent field and matter. The sensitivity of micro- ring resonant sensors made from strip waveguides is generally around 100 nm/RIU [32], which means that the refractive index of the cladding changes by 0.001 and the position of the resonant peak changes by only 0.1 nm. This is because the optical field of the strip waveguide is basically limited in high refractive index media, manifested as weak interactions between environmental substances and evanescent fields. To further improve sensitivity, several waveguides have been proposed and demonstrated. Sigle slot waveguide, double slot waveguide, Bragg grating waveguide, thin waveguide has been proposed as microring resonators refractive index sensors to achieve sensitivities of 298 nm/RIU [33], 476 nm/RIU [34], 563 nm/RIU [35], 297 nm/RIU [36] and 133 nm/RIU [37], respectively.

Subwavelength grating (SWG) waveguide [38], have become important tools in integrated photonics due to their design flexibility and precise photolithography control over the effective refractive index and mode field distribution of waveguides [39,40]. Compared with traditional microring resonant refractive index sensors, ring resonators based on SWG have higher sensitivity, owing to the strong limitation of the electric field in SWG waveguides in the low refractive index region between high refractive index dielectric blocks, which is inaccessible in traditional band or slot waveguides. Since its first demonstration as a refractive index sensor, subwavelength grating waveguide microring (SWGMRR) resonators have attracted strong research interest because of their unique electric field distribution [41,42,43,44]. Subsequently, a sensitivity of 490 nm/RIU [45] and a Q factor of 7000 were achieved through a SWGMRR. By further optimizing the geometric parameters of the SWG waveguide, the sensitivity of 664 nm/RIU [46] was obtained, and the Q factor was increased to 15,918 by using a retrack SWGMRR. A single slot SWGMRR was proposed, which minimizes the propagation loss of the optical field by designing the inner ring subwavelength grating waveguide block into a trapezoidal shape. At the same time, the groove distribution of the subwavelength grating waveguide block greatly improves the interaction between the optical field and the analyte, ultimately achieving ultra-high sensitivity of 823 nm/RIU [47] and a Q factor of up to 25,000. From the above, subwavelength grating waveguides have excellent performance and potential in refractive index sensing. On the one hand, improving the existing silicon photonic microring refractive index sensors is a common goal of researchers, hoping to further achieve detection performance such as SPR [7]. On the other hand, microring refractive index sensors with excellent sensitivity are highly anticipated in trace analyte detection, and have great potential applications in portable acute disease diagnostic sensors [48]. Given the above viewpoints, we propose a microring refractive index sensor with high sensitivity and low limit of detection.

In this paper, to further enhance the interaction between the light field and the analyte to improve sensitivity, we propose a microring resonator refractive index sensor based on a hybrid waveguide block double slot subwavelength grating (HDSSWG) waveguide. By carefully optimizing the structure and parameters of the device, a sensitivity of up to 1005 nm/RIU has been achieved, which is 182 nm/RIU higher than the reported SWG refractive index sensor based on silicon optical technology. Furthermore, by carefully simulating the shapes of several waveguide blocks to reduce waveguide losses, a Q factor of 22,429 has been achieved. The minimum gap size between the waveguide blocks of our proposed sensor is 90 nm, which can be fabricated on a standard silicon photonics processing platform.

## 2. Design and Optimization of Device

The structure of the double slot subwavelength grating waveguide microring resonator (DSSWGMRR) is illustrated in Figure 1a, consisting of a straight coupled double slot subwavelength grating waveguide and a circular HDWSWG waveguide with an outer radius of R. The coupling gap, the grating period, the rectangle width, the middle rectangle length, the middle rectangle width, slot width, the trapezoidal length are labeled as G, Ʌ, L_s_, W_s_, L_m_, W_m_, W_g_ and W_t_, respectively. The water solution with different refractive index concentrations represented by the light blue coating solution is used as the sensing medium; The material used is silicon on insulator (SOI), with a thickness of the buried oxygen layer and top layer silicon, respectively 2 μm and 220 nm, as shown in Figure 1b.

When λ/Λ >> 2n_eff_ is satisfied, the subwavelength grating waveguide operates in the subwavelength transmission region, which does not diffract light in the far field, and propagates Bloch mode like the conventional strip waveguide. In order to select the appropriate period, we used the 2D Finite Difference Eigenmode (FDE) to analyze the effective refractive index and group refractive index (in the paper, TE fundamental mode is used) of the transmission mode in DSWWG, which can be calculated as below [49]:(1)$${n_{eff}}^{2}={fn_{effsi}}^{2}+(1-f){n_{effc}}^{2},$$
(2)$$n_{g}=n_{eff}-\lambda \frac{\partial n_{eff}}{\partial \lambda },$$
where n_eff_ and n_g_ are the effective refractive indices of subwavelength grating waveguides, f(=W_s_/Λ) is the duty cycle, equal to the ratio of waveguide block width to period, n_effsi_ is the refractive index of silicon, and n_effc_ is the effective refractive index of the cladding medium. Considering the manufacturing line width, 300 nm was chosen. As shown in Figure 2a, the effective refractive index and group refractive index of DSSWG with four different parameters (Taking the first type as an example, “0.1”, “0.22”, and “0.5” respectively represent the width of the slot W_g_ is 100 nm, the width of the silicon pillar L_s_ is 220 nm, and the duty cycle is 0.5) at 300 nm period are depicted.

The electric field distribution of HDSWWG was calculated by using a full 3D vectorial Finite-Difference-time-Domain (3D-FDTD) approach. Figure 2b–d shows the electric field distribution along the propagation axis (*x*-axis) for a DSSWG waveguide with L_s_ = L_m_ = 220 nm, W_t_ = W_m_ = W_s_ = 220 nm, period Λ = 300 nm, duty cycle f = 0.5, and λ = 1550 nm. Figure 2a,c show the distribution of electric field intensity at z/2 (x-y profile), the center section of the silicon waveguide block (y-z plane), and the center section of the space (y-z plane), respectively. It is obvious that the electric field is mainly distributed in the low refractive index region at the gap, which is the reason why DSWWG has high sensitivity. At the same time, more electric field intensity is distributed in the middle silicon waveguide block, which is obtained through more detailed observation.

To better illustrate the design process, several objective indicators used to evaluate sensor performance need to be defined, including quality factor(Q), bulk sensitivity (S_b_), surface sensitivity (S_s_), and the limit of detection (LoD), which limit of detection is divided into intrinsic detection limit (iLoD) and system detection limit (sLoD). The physical meaning of the Q factor is a transient response phenomenon, which represents the number of roundtrip times when the energy of the circulating photons in the ring decays to 1/e of the initial energy. From this perspective, reducing the loss of the ring is a key factor in improving the Q factor [50]:(3)$$Q=\omega_{res}\frac{\varepsilon }{\partial \varepsilon /\partial \tau }=\frac{4.34\cdot 2\pi n_{g}}{\lambda_{res}\alpha (dB/m)},$$

Intuitively, the Q factor represents the sharpness of the resonant peak of the micro ring resonator, which can be calculated as the ratio of the resonance wavelength (λ_res_) and full width at half maximum (FWHM) of the transmission spectrum as the expression [50]:(4)$$Q=\frac{\lambda_{res}}{\Delta \lambda_{FWHM}},$$
where ω_res_ is the resonant angular frequency, ng is the group refractive index of the propagation mode, ∂ε/∂τ is the energy transition time of the propagation mode, Δλ_FWHM_ is the full width at half maximum of the resonant peak.

In silicon photonics refractive index biosensors, two types of sensitivity are used as metrics. The first type is volume refractive index sensitivity, which is defined as [50]:(5)$$S_{b}=\frac{\lambda_{res}}{n_{g}}\frac{\partial n_{eff}}{\partial n_{c}}=\frac{\Delta \lambda_{res}}{\Delta n_{c}},$$
where λ_res_ and ng are the group refractive indices of the resonant peak wavelength and propagation mode, respectively, the volume refractive index refers to the change in the position of the resonant peak (“Δλ_res_”) caused by a change in the refractive index of a unit cladding medium(Δn_c_), in units of (nm/RIU).

Similarly, surface sensitivity is expressed using the following [50]:(6)$$S_{b}=\frac{\lambda_{res}}{n_{g}}\frac{\partial n_{eff}}{\partial t_{ad}}=\frac{\Delta \lambda_{res}}{\Delta t_{ad}},$$

Surface sensitivity is used to detect the change in resonance wavelength (∆λ_res_) resulting from the change of the adsorbed molecule layer thickness which can change the refractive index above the device, in units of (nm/nm). For most proteins, the refractive index is about n_ad_ = 1.48.

The system limit of detection(sLoD) is another widely used parameter used to evaluate the sensing capability of sensors, and is defined as the resonance wavelength resolution 3σ divided by the sensitivity (S_s_ or S_b_), where σthe standard deviation of the noise causing the resonance peak position to shift, which can be defined as the minimum cladding refractive index or minimum cladding medium thickness change to induce the detectable change in the resonance peak position or output signal [39]:(7)$$sLoD=\frac{3\sigma }{\left(S_{s}\;or\;S_{b}\right)},$$

However, it is difficult to objectively compare the performance of sensors characterized by different experimental testing conditions and experimental systems, so the intrinsic limit of detection is introduced as another representation [40]:(8)$$iLoD=\frac{3\sigma }{\left(S_{s}\;or\;S_{b}\right)}$$
where λ_res_ is the resonant wavelength of the resonator, Q is the quality factor of the resonator. iLoD can be converted into the minimum refractive index change required to shift the resonant wavelength by unit linewidth.

To achieve the best sensing performance, firstly, the sensitivity and loss characteristics of standard rectangular block straight double slot subwavelength grating waveguides were studied. Here, the duty cycle f and period Λ are fixed at 50% and 300 nm, and the waveguide block width L_s_ is scanned for different slot widths W_g_. The sensitivity of standard DSWWG was analyzed using FDE. Figure 3a shows the sensitivity curve obtained by scanning the waveguide width L_s at different gap widths. As the waveguide width decreases and the gap width increases, a nonlinear increase in sensitivity can be observed. As shown in Figure 3b, the losses corresponding to the TE fundamental mode (the mode used in this paper) with a propagation length of 100 grating periods Λ were obtained through 3D-FDTD simulation, and the structure parameters were set with the same sensitivity. Obviously, smaller L_s_ and slot widths W_g_ can lead to excessive losses, thereby reducing the Q factor. However, compared to the reduced sensitivity caused by a smaller waveguide width, choosing a larger slot width to achieve high sensitivity is a better choice. This can be observed around 900 nm/RIU, and increasing the slot width cause minimal loss increase while significantly improving sensitivity. Considering the potential for further improvement in sensitivity and loss, a rectangular block length of 180 nm has been determined as the next design step. Considering the dependence on duty cycle f, L_s_ is fixed in180 nm after weighing sensitivity and loss, and three slot widths W_ g = 100 nm, 120 nm, and 140 nm were selected for designing the duty cycle. The relationship between sensitivity, loss, and duty cycle is shown in Figure 3c,d. Similarly, a smaller duty cycle leads to greater sensitivity and loss. However, the loss at 120 nm W_g_ is greater than at 140 nm W_g_. We extend the single wavelength transmittance to broadband, and a wavy curve is observed, which is due to the high refractive index waveguide block acting as the emitting mirror to form the ‘F-P’ cavity. Essentially, the increase in sensitivity and loss is caused by reducing the limitation of waveguides on the optical field and increasing the interaction between the optical field and the environmental medium.

Based on the above analysis, although a high sensitivity can be achieved through a very small waveguide width and duty cycle, or a very large slot width, the high sensitivity brought about by this method also results in an unequal loss, which leads to an unacceptable low Q factor in sensing measurements. Therefore, we discussed several deformation structures of DSSWG to achieve high sensitivity while reducing losses. Among them, the most cost-effective structure is used for simulating the final transmission spectrum and sensing characteristics. Firstly, based on the results of the first round of analysis, the basic double slot subwavelength grating waveguide with a waveguide width of 180 nm, a slot width of 140 nm, and a 40% duty cycle was selected as the basis for deformation. Meanwhile, considering bending losses, a bending radius of 30 μm is selected. As shown in the analysis of the electric field intensity distribution in Figure 2, more electromagnetic fields are confined within the middle silicon waveguide block. Therefore, reducing the size of the middle silicon waveguide block is a method to improve sensitivity. Here, simulation results were obtained by using 2.5D-FDTD simulation, where the mesh size is set to 4, the input source, the wavelength range and simulation time are set as fundamental TE fundamental mode source, 1500 nm to 1600 nm and 250,000 fs, respectively.

The ordinary straight double groove subwavelength grating waveguide don’t achieve the expected sensing effect, so we studied the sensing characteristics of a de-formed double groove subwavelength grating waveguide with a smallsized waveguide block in the middle and symmetrical trapezoidal waveguide blocks on both sides. Firstly, the size of the intermediate waveguide block is reduced to increase sensitivity, and then W_t is increased to increase the Q factor while slightly reducing sensitivity. Figure 4a illustrates the Q factor and sensitivity trend corresponding to the reduction of the intermediate silicon block, where fm = L_m_/L_s_ = W_m_/W_s_. As the fm decreases, the sensitivity is improved and the Q factor is reduced. However, near the sensitivity of 1000 nm/RIU, the Q factor is only around 13,000, indicating a low detection limit of 10^−4^ orders of magnitude. Based on fm = 0.5, which sensitivity and Q factor are 1023 nm/RIU and 5780, respectively, we investigated the changing trends of Q factor and sensitivity under different W_t_. In Figure 4b, the phenomenon of slow decrease in sensitivity and significant increase in Q factor is observed. Especially at W_t_ = 210 nm, a 1005 nm/RIU and sensitivity, as well as a 22,429 Q factor, were displayed. Compared to the initial state (W_t_ = 120 nm), the sensitivity only decreased by 18 nm/RIU while the Q factor increased by about twice. Considering the difficulty of device fabrication and the tradeoff between sensitivity enhancement and sensitivity degradation with W_t_ increasing, W_t_ is set to 210 nm instead of 240 nm. The entire simulation result proves the correctness of the initial idea.

Finally, the influence of manufacturing errors was considered, which is partly due to the roughness of the waveguide sidewall. Which rough sidewall will lead to scattering loss occupying the main advantage in the total optical transmission loss, thereby reducing the Q factor, which is an inherent influence of the process. On the other hand, it is the error caused by geometric dimensions deviating from the design, which leads to changes in the output and sensing characteristics of the device. In the previous analysis, the size of the waveguide block varies in steps of 30 nm, so the impact of size error of +/−10 nm on the device is slight. For standard silicon photonic foundries, the processing error of +/−10 nm can be easily maintained. In summary, the devices we propose exhibit insensitivity to machining errors.

To further optimize the sensing performance of the sensor, it is necessary to maximize the extinction ratio (representing the ratio of the minimum to maximum value of the resonant peak) and Q factor of the target resonant peak. These parameters can be optimized by scanning the coupling gap. As shown in Figure 5a,b, at a coupling gap of 1250nm, the extinction ratio and Q factor near the 1550nm resonant peak reach their maximum values simultaneously, which is beneficial for sensing measurement. Meanwhile, the relationship between the sensitivity changes of microrings under different coupling gaps has been studied. As shown in Figure 5c, near the optimal coupling gap, there is little fluctuation in sensitivity, which means that the sensor can resist changes in coupling distance caused by protein molecule filling and manufacturing errors.

## 3. Results

The complete spectrum was obtained by using 2.5D-FDTD simulation, where the mesh size is set to 4, the light source, the wavelength range and simulation time are set as fundamental TE mode source, 1500 nm to 1600 nm and 250,000 fs, respectively. Figure 6 shows the initial normalized transmission spectrum of the sensing device exposed under the deionized water layer, with three resonant peaks located at 1539.74 nm, 1547.59 nm, and 1555.52 nm, respectively. The illustration shows a full width at half maximum of 0.069 nm at the resonant peak of 1547.59 nm, and the calculated Q factor is 22,429. A 24.73 dB ER was observed, which corresponds to the result of a 1.25 μm coupling gap. The electric field distribution of the HDSSWG-MRR at the resonant wavelength is in Figure 7b. Figure 7b and Figure 7c respectively display the distribution of electric field intensity in the y-z cross section at the center of the silicon block and the local magnified view of the microring annular section. The electric field strength is mostly limited to the groove, and the external electric field extends a long-distance outward, which is more conducive to biological modification sensing.

In order to demonstrate the sensing characteristics of the proposed device in more detail, the cladding with different refractive indices was modified to characterize the volume sensitivity. For surface sensitivity, the dual characteristics of biology and devices are tested. On the biological side, including the selected biomaterials and surface modification methods, which means that experimental measurements are the best form of expression. Therefore, here, volumetric sensitivity is chosen to display sensitivity.

The transmission spectrum of HDSSWG-MRR refractive index sensors in different refractive index cladding are shown in Figure 8a, where the initial resonant wavelength at 1.33300 is 1547.595 nm. By increasing the refractive index of the solution to 1.33425, 1.33550, 1.33625, 1.33800 and 1.33925, the device spectrum is shifted to 1548.851 nm, 1550.108 nm, 1551.364 nm, 1552.621 nm and 1553.877 nm. This clearly demonstrates that even small changes in the refractive index of the analyte can cause significant shifts in the resonance peak wavelength. According to calculations, the refractive sensitivity of the device is 1005 nm/RIU and the LoD is 6.86 × 10^−5^ RIU. The sensitivity of the displayed silicon refractive index sensor is 1005 nm/RIU, which is higher than the reported refractive index sensor. By using a hybrid double slot subwavelength grating waveguide with trapezoidal columns as the inner and outer layers and small rectangular waveguide blocks as the middle layer, compared with the waveguide structure corresponding to the same sensitivity, the Q factor has been greatly improved, which is attributed to the unique waveguide structure and variation process. Although the bending radius of 30 μm introduces a small free spectrum range (FSR) of about 8 nm and integration issues. However, in practical sensing applications, this is more in line with the width of microfluidic channels.

On the other hand, a microring resonators corresponds to a specific biological modification layer, which means that a single channel corresponds to a single substance detection. From these aspects, the size increase problem caused by a radius of 30 μm is not as strict as in communication applications. Finally, an 8 nm FSR is sufficient for biosensing applications in the order of pmol/fmol [51,52,53,54].

The performance of recent SOI MRR-based refractive index sensors is shown in Table 1. The proposed HDSSWG-MRR achieves the highest sensitivity of 1005 nm/RIU to date under the refractive index of the water stop solution, while the sensitivity of most resonant refractive index sensors is less than 500 nm/RIU. Compared with the reported subwavelength grating waveguide refractive index sensors of 664 nm/RIU [46] and 823 nm/RIU [47], the sensitivity has been improved by 341 nm/RIU and 182 nm/RIU, respectively, which represents a leap in sensitivity. On the other hand, the designed structure does not require a particularly high machining accuracy. The minimum size between waveguide blocks in the ring is 90 nm, and the maximum is 150 nm, which is a very acceptable size for silicon optical foundries. Although the Q factor is 22,429, which does not reach a highest value, in terms of the practical application of existing silicon refractive index sensors in biological material detection, the contact area between the organism and the modified layer and the detected object, or the system sensitivity, may be a more important characteristic for achieving extremely low detection limits [55,56,57,58]. It is not simply judged by the Q factor that our sensor exhibits the physical properties of constructing waveguides. In addition, the small full width at half height caused by the ultra-high Q factor can lead to unpredictable errors in experimental data measurement and harmonic curve fitting.

## 4. Conclusions

In summary, we propose a microring resonant refractive index sensor based on a hybrid dual slot subwavelength grating waveguide in silicon photonics technology. The optimal sensing performance is achieved by optimizing the shape and structure of the waveguide. Achieved a volume sensitivity of up to 1005 nm/RIU, with calculated Q factors and intrinsic detection limits of 22,429 and 6.86 × 10^−5^ RIU, respectively. The key innovation lies in the design of a unique and novel waveguide structure that achieves ultra-high sensitivity, which, to our knowledge, is currently the maximum value that can be achieved through a single microring resonator under the refractive index of aqueous solutions. Compared with ordinary rectangular block waveguide structures, colleagues have reduced bending losses. The designed silicon photon refractive index sensor has a large contact area and great potential for application in biosensing based on chip laboratory detection.

## Figures and Tables

**Figure 1 sensors-24-01929-f001:**
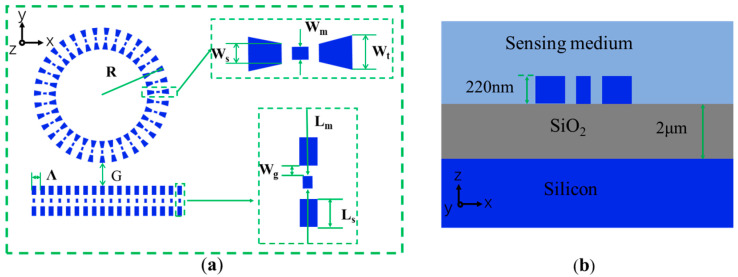
(**a**) Schematic of a hybrid waveguide block double slot subwavelength grating (HDSSWG) waveguide resonator and design parameters of HDSSWG waveguide block. (**b**) The magnified waveguide cross section exposing in a sensing medium. The model is not in scale.

**Figure 2 sensors-24-01929-f002:**
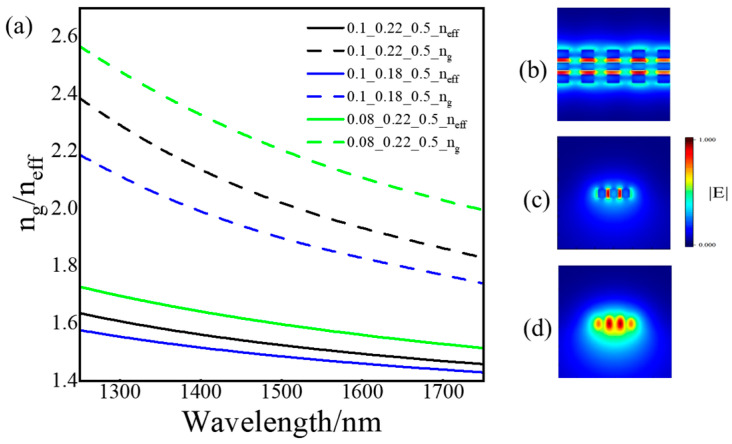
(**a**) The effective refractive index and group refractive index of DSSWG with four different parameters; (**b**–**d**): Electric field magnitude distribution in the x-y plane defined by a cut at z/2 for a DSSWG waveguide with dimensions of L_s_ = L_m_ = 220 nm, W_t_ = W_m_ = W_s_ = 220 nm, period Λ = 300 nm, and λ = 1550 nm; Distribution of the z-component; (**c**) Distribution of the y-z plane in the middle of Si block, and (**d**) Cross-section in the middle of the gap.

**Figure 3 sensors-24-01929-f003:**
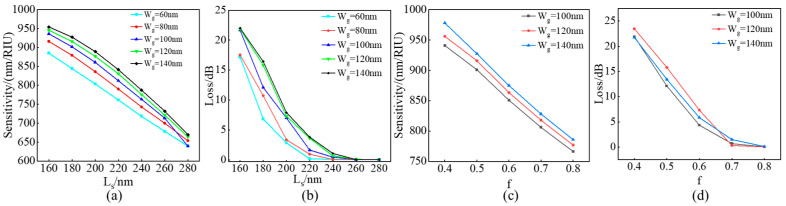
(**a**,**b**) Sensitivity and loss (100 grating periods length) as a function of silicon waveguide block length at 60 nm, 80 nm, 100 nm, 120 nm, 140 nm slot width. (**c**,**d**) Sensitivity and loss (100 grating periods length) as a function of duty cycle at 100 nm, 120 nm, 140 nm slot width.

**Figure 4 sensors-24-01929-f004:**
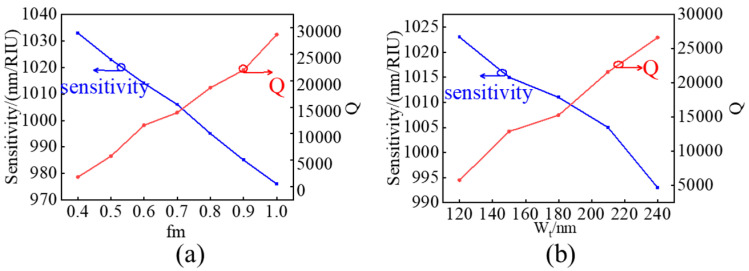
(**a**) Sensitivity and Q factor under different fm. (**b**) Sensitivity and Q factor as under different W_t_.

**Figure 5 sensors-24-01929-f005:**
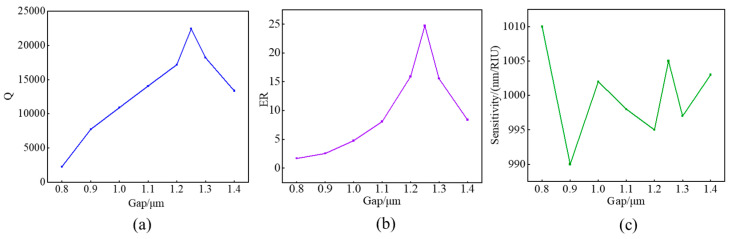
(**a**–**c**) Q, ER and sensitivity of the HDSSWG-MRR under different coupling gaps.

**Figure 6 sensors-24-01929-f006:**
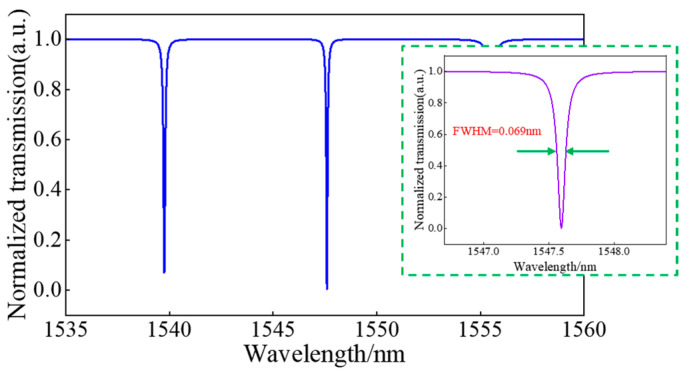
Normalized transmission spectrum of the HDSSWG-MRR with the deionized water cladding. The illustration in the green dashed box shows the full width at half height of the main resonant peak more clearly.

**Figure 7 sensors-24-01929-f007:**
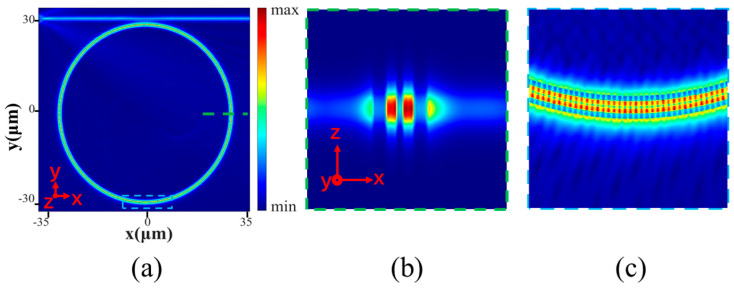
(**a**) Electric profile of the HDSSWG-MRR at the resonant wavelength. (**b**) Electric field magnitude distribution in the y-z in the center of silicon block. (**c**) Electric field magnitude distribution in the x-y plane defined by a cut at z/2.

**Figure 8 sensors-24-01929-f008:**
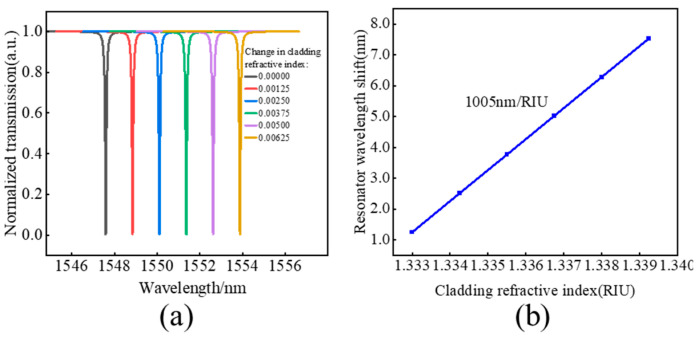
(**a**) The transmission spectra of the SWGRMR refractive index sensor in the different refractive index, and (**b**) The sensing performance of the single SWGRMR sensor.

**Table 1 sensors-24-01929-t001:** Performance comparison of recent microring refractive index sensors with different types of waveguides.

Device Structure	Bulk Sensitivity (nm/RIU)	Q	LoD	Ref.
Ultra-Thin waveguide MRR	133	15,000	5 × 10^−4^	[25]
R-SWGSMRR	664	15,918	1.43 × 10^−4^	[33]
T-SSWGMRR	823	24,941	7.53 × 10^−5^	[34]
Multi-Box SWGMRR	563	2600	1.02 × 10^−3^	[37]
Double slot MRR	576	16,000	3.7 × 10^−6^	[23] *
SWGSMRR	490	7300	2 × 10^−6^	[31] *
R-SWGSMRR	429.7	9800	3.71 × 10^−4^	[1]
T-SWGMRR	440.5	9100	3.9 × 10^−4^	[2]
Single Slot MRR	476	1900	2.1 × 10^−6^	[22] *
SWGMRR	383	4000	N/A	[28]
HDSSWG-MRR	1005	22,429	6.86 × 10^−5^	This Work

The reference with * is the sLoD.

## Data Availability

The data analyzed during the current study are available from the corresponding author upon reasonable request.

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
