# Peer review of "Ultrasensitive Silicon Photonic Refractive Index Sensor Based on Hybrid Double Slot Subwavelength Grating Microring Resonator"

_sensors, 2024, doi:10.3390/s24061929_

Round 1

Reviewer 1 Report

Comments and Suggestions for Authors

The authors utilized the hybrid double slot subwavelength grating microring  resonator for silicon photonic refractive index sensor. This might be publishable if the authors addressed some of the concerns:

In their results they indicated that the proposed HDSSWG-MRR achieves the highest sensitivity of 1005nm/RIU to date 327 under the refractive index of the water stop solution, while the sensitivity of most resonant 328 refractive index sensors is less than 500nm/RIU. However, there are researches that has higher than what is mentioned like:

1)    an  S-shaped double-spiral microresonator which has a bulk sensitivity of  1410 nm/RIU when detecting 1 µL IPA solutions inside a 200 µm wide microchannel (DOI: 10.3390/s23136177)

2)    A nested micro-ring sensor based on a subwavelength grating waveguide and the Vernier effect achieved a sensitivity of 8030 nm/RIU near a 1550 nm wavelength in a deionized water environment (DOI: 10.1364/AO.496107)

In addition, the authors need also to address the following:

1)    In Fig. 3b, why is it that at the 60 and 100 nm block length have a bigger drop from 160 to 180 nm waveguide width which was not observed with other block lengths.

2)    In Fig 3d, please expand explanation with 120 has a  higher loss than 140

3)    In Fig 5c, the authors indicate that sensitivity is insensitive to changes in coupling 248 spacing, they need to present a more scientific way to determine insensitivity, it seem sthat at 0.8 to 0.9 gap has higher and lowest sensitivity

4)    For table 1, how many samples were tested, the authors need to properly present significant figures like 1g/100 mL is 1.33425 and 0g/100 mL is 1.333 is it 1.33300 or other digits

Reviewer 2 Report

Comments and Suggestions for Authors

Dear Authors and Editor!

Thank you for the possibility to evaluate this interesting work!

In my opinion, following questions / concerns should be addressed:

1) The model includes a huge number of design parameters (such as W_g, L_s, etc.) introduced in Fig.1. However, the discussion on the choice of these parameters is scattered throughout the entire manuscript. It appears necessary to state explicitly (within a single section) which of the above parameters are varied in the work and which are fixed (also providing certain comments on the criteria to choose the values used).

2) Comments on the Figures:

- Fig.4 appears somewhat confusing; nearly identical images correspond to the optimization of the waveguide parameters (fm) and micro-ring parameters (W_t), which are not logically linked.

- In Fig.6, panel (a), given in linear scale, appears redundant. The inset is informative, but it can be allocated in the panel (b).

- In Fig.7, panel (a) occupies too much space so that panels (b) and (c) – the most interesting ones – are barely visible. I suggest rearranging the figure.

3) Since the work is purely theoretical, certain comments should be given on its possible experimental verification. Namely, is the presented scheme sensitive to the fabrication errors (since lithography and etching steps may somewhat distort the original design)?

As a general comment, it is sometimes hard to follow the line of thought in the manuscript. It might be partially related to a large number of parameters the Authors use in their calculations, which are not appropriately designated. In my opinion, parameter names should be intuitively understandable and clearly separated for the waveguide and ring parts of the circuit.

Though I am not a specialist in the area of photonic micro-circuits, I believe that presented novel design indeed offers significant progress in sensing efficiency, and thus the work may be published in “Sensors” after a minor revision.

Comments on the Quality of English Language

Though minor language issues are present in the manuscript, it is completely undestandable.

Round 2

Reviewer 1 Report

Comments and Suggestions for Authors

The authors have satisfactorily addressed the concerns/suggestions made by the reviewers thus it is now acceptable for publication.